# Clinical Trials in Prader–Willi Syndrome: A Review

**DOI:** 10.3390/ijms24032150

**Published:** 2023-01-21

**Authors:** Ranim Mahmoud, Virginia Kimonis, Merlin G. Butler

**Affiliations:** 1Department of Pediatrics, University of California, Irvine, CA 92697, USA; 2Department of Pediatrics, Faculty of Medicine, Mansoura University, Mansoura, 35516, Egypt; 3Departments of Neurology and Pathology, University of California, Irvine, CA 92697, USA; 4Children’s Hospital of Orange County, Orange, CA 92868, USA; 5Departments of Psychiatry & Behavioral Sciences and Pediatrics, University of Kansas Medical Center, Kansas City, KS 66160, USA

**Keywords:** Prader–Willi syndrome, obesity, hyperphagia, clinical trials, genetics

## Abstract

Prader–Willi syndrome (PWS) is a complex, genetic, neurodevelopmental disorder. PWS has three molecular genetic classes. The most common defect is due to a paternal 15q11-q13 deletion observed in about 60% of individuals. This is followed by maternal disomy 15 (both 15 s from the mother), found in approximately 35% of cases. the remaining individuals have a defect of the imprinting center that controls the activity of imprinted genes on chromosome 15. Mild cognitive impairment and behavior problems in PWS include self-injury, anxiety, compulsions, and outbursts in childhood, impacted by genetic subtypes. Food seeking and hyperphagia can lead to morbid obesity and contribute to diabetes and cardiovascular or orthopedic problems. The control of hyperphagia and improving food-related behaviors are the most important unmet needs in PWS and could be addressed with the development of a new therapeutic agent, as currently no approved therapeutics exist for PWS treatment. The status of clinical trials with existing results for the management of obesity and hyperphagia in PWS will be discussed in this review, including treatments such as beloranib, setmelanotide, a diazoxide choline controlled-release tablet (DCCR), an unacylated ghrelin analogue, oxytocin and related compounds, glucagon-like peptide 1 receptor agonists, surgical intervention, and transcranial direct-current stimulation.

## 1. Introduction

Prader–Willi syndrome (PWS) is a rare, genetic, neurodevelopmental disorder caused by errors in a complex genomic mechanism, referred to as genomic imprinting, comprised of three PWS molecular genetic classes. PWS is characterized by severe hypotonia with a poor suck and feeding difficulties causing failure to thrive during infancy, hypogenitalism/hypogonadism in both sexes, motor and cognitive delays, low muscle tone, slow metabolism, behavior disturbances, and endocrine findings involving growth and other hormone deficiencies with short stature, infertility, and small hands and feet (e.g., [1,2,3,4,5,6]). Mild cognitive impairment and behavior problems, including self-injury, anxiety, compulsions, and outbursts, can occur in childhood along with food seeking and hyperphagia, leading to morbid obesity and a shortened life expectancy if not controlled (e.g., [7]). PWS is considered the most commonly known cause of life-threatening obesity in humans, affecting approximately 400,000 people worldwide and one in every 20,000 live births [8]. Occurrences of this rare disorder are sporadic, but the cause is due to errors in the genomic imprinting of the chromosome 15q11-q13 region in humans. The most common defect is due to a paternal 15q11-q13 deletion observed in about 60% of individuals. This is followed by maternal disomy 15 (both 15 s from the mother), found in approximately 35% of cases. The remaining individuals have a defect of the imprinting center that controls the activity of imprinted genes on chromosome 15 or have chromosome 15 translocations or inversions (e.g., [4]). Individuals with the different PWS molecular classes present with varying clinical findings. Those with the typical 15q11-q13 deletion, specifically the larger 15q11-q13 type I deletion, have a more severe phenotype with self-injuries, compulsions, and lower cognition than those with the typical, smaller 15q11-q13 Type II deletion or maternal disomy 15 (e.g., [9,10,11]). A typical patient with Prader–Willi syndrome is shown in Figure 1.

If not externally controlled, significant hyperphagia leads to morbid obesity in PWS and contributes to diabetes, cardiovascular or orthopedic problems, and even death [12]. The most common cause of death in PWS is respiratory failure followed by cardiac failure, gastrointestinal failure, and infection [7,13]. According to a 2014 survey of parents and caregivers of PWS patients, reducing hunger and improving food-related behaviors were the most important unmet needs in PWS. These needs could be addressed with the development of a new therapeutic agent, as currently no approved therapeutics exist for the treatment of hyperphagia in PWS.

## 2. Beloranib Clinical Trial in Prader–Willi Syndrome

Beloranib treatment was undertaken in a large cohort of individuals with genetic confirmation of PWS in a clinical trial sponsored by Zafgen, Inc. (Boston, MA, USA). The goal was to investigate the efficacy of beloranib treatment for hyperphagia and obesity, as well as to determine its safety with tolerability over 26 weeks of treatment in both PWS adolescent and adult participants. The study was stimulated by early reported data indicating a clinically significant and sustained weight loss with decreased hunger in obese subjects using beloranib [15]. Beloranib inhibits methionine aminopeptidase 2 (MetAP2) by removing methionine residue from proteins, impacting fat metabolism. Inhibitors of MetAP2 were found to reduce food intake, body weight, fat content and adipocyte size in animal models [15]. Hence, a Phase Three, randomized, placebo-controlled, double-blind trial was conducted in 15 states in the United States between 2014 and 2015. Those with genetically confirmed PWS were enrolled between the ages of 12 to 65 years. Participants had an elevated body mass index (BMI) (ages 12 to 17 years: BMI ≥ 95th percentile for age and sex; ages 18 to 65 years: BMI 27–60 kg/m^2^), with a total score ≥ 13 on the Hyperphagia Questionnaire for Clinical Trials (HQ-CT), and a stable weight for at least three months, but otherwise demonstrated health-related findings, such as blood pressure readings, within the normal range. Those with type II diabetes were accepted, if stable. Growth hormone treatment was allowed if a stable dose was prescribed for at least three months before entry into the trial. Participants living in a group home for less than 50% of the time were excluded. All participants were randomized via computer access using a centralized interactive system with a 1:1:2:2 ratio of lower dose placebo: higher dose placebo: lower dose beloranib (1.8 mg): higher dose beloranib (2.4 mg) per participant.

Following a two-week single-blind placebo lead-in, participants were randomized to study treatment with doses selected based on previous experience, with the study drug administered twice weekly. The study included an optional twenty-six-week, open-label extension trial in which all participants received four weeks of 1.8 mg beloranib, followed by 2.4 mg beloranib use for 22 weeks. The prespecified co-primary endpoints were a change in hyperphagia-related behaviors, captured via a questionnaire form, and a percent change in body weight from baseline to week 26. The HQ-CT form consisted of nine question items with responses ranging from 0 to 4 units each, with a possible total score range of 0 to 36 designed to measure symptoms of food-related preoccupations and behaviors. The form was completed by the caregiver for each participant. It was estimated that a sample size of 20 participants per group would provide 94% power to detect the between-group difference in each of the co-primary endpoints. Out of 126 screened individuals, 108 participants were randomized and 107 received the study drug. Demographic and baseline characteristics were well-matched across the treatment groups for age, sex, growth hormone use, weight, BMI, fat mass, and genetic subtype. The participants were primarily Caucasian. A significant reduction in fat mass was found in both the low- and high-dosage beloranib-treated groups when compared with the placebo, with participants having a weight loss greater than or equal to 5%. A change in HQ-CT individual question scores indicating improvement were found for eight out of the nine measures in the beloranib-treated participants. The questionnaire measures were classified as: upset when denied food, bargain to get more food, forage through trash for food, up at night to seek food, persistent asking for food, time spent talking about food, distress when stopped from asking about food, and interferes with activities. Unfortunately, the randomized, double-blind portion of the trial was stopped early due to venous thromboembolic events, including two participant deaths in the beloranib-treated group. 

## 3. Oxytocin Clinical Trials in PWS

Oxytocin is a neuropeptide hormone which plays an important role in social interactions, social skills, food intake, anxiety, energy expenditure, maternal behaviors, and body weight regulation [16,17]. All of these parameters are severely affected in patients with PWS. They have a decreased number of oxytocin-producing neurons in the hypothalamic periventricular nuclei and a small number of observed periventricular nuclei [18]. The deficiency of these neurons could be related to PWS patients’ poor social judgement and their inability to control their emotions. Two mouse models deficient for PWS genes, including necdin and Magel2, also showed decreased oxytocin levels [19]. Additionally, postnatal treatment with oxytocin has been reported to normalize suckling and feeding behavior in the Magel2-knockout mouse [20]. Approximately one half of the surviving mice showed an improvement in social recognition skills, which would suggest that oxytocin could be helpful in managing behavioral problems in PWS [21].

To date, there have been seven clinical trials on the use of oxytocin in PWS. The first study was a double-blind, randomized, placebo-controlled trial undertaken by Tauber et al. [22] in 2011. Twenty-four patients were included, with a median age of 28.5 years, and grouped based on gender and intelligence quotient (IQ). Every patient received a single dose (24 IU) of either oxytocin or a placebo. The behavior of the patients was assessed three times: the first at two days before drug administration, the second at one half-day after administration (early effects), and for the last time at two days after drug administration (late effects). A behavior grid was developed to assess ten social and emotional behaviors. There was also as a separate set of four questions relating to eating behavior. There was no difference between the two groups before intranasal administration, but a significant increase in trust in others, less sadness tendencies, and disruptive behavior were observed in the two days after intranasal drug administration. While no statistical difference was observed in the eating behavior scores between both groups, there was improvement in social skills in the oxytocin-treated group. The limitations of this study were the use of only one dose of oxytocin and a placebo with the effect of treatment appearing only in the behavior grid.

The second study was a double-blind, randomized, controlled trial with intranasal use of oxytocin in patients with PWS to examine its effect on physical, behavioral, and cognitive function. Thirty patients with PWS were included in this study, and two different doses of oxytocin were used according to the age of patients: a 24 IU dosage for patients aged 16 years and more, and an 18 IU dosage for patients aged between 12 and 15 years. The dose was increased to 40 IU for patients older than 16 years and 32 IU for patients between 12 and 15 years. The oxytocin and placebo were given for eight weeks; this was followed by a two-week washout period. The Developmental Behavior Checklist (DBC), the Yale–Brown Obsessive Compulsive Scale (Y-BOCS), the Dykens Hyperphagia Questionnaire, the Reading the Mind in the Eyes Test (RMET), and the Epworth Sleepiness Scale (ESS) were used to assess behavioral and related problems. No improvement was found in any of the behavioral problems or in hyperphagia in response to the oxytocin treatment [23].

The third trial was applied only to children with PWS [24]. The randomized, double-blind, placebo-controlled, cross-over study was conducted on 25 children with the aim of investigating the effect of oxytocin on social and eating behavior in PWS. Children received either oxytocin or a placebo for four weeks. They then crossed over to the alternative treatment for another four weeks. The Dykens Hyperphagia Questionnaire was used to evaluate changes in eating behavior and hyperphagia and the Oxytocin Study Questionnaire, designed by PWS experts, was used for the evaluation of social behavior. When the participants were divided into two groups, <11 years and >11 years, there was no significant effect between either oxytocin or the placebo in the whole group. The younger children showed beneficial effects of oxytocin on social behavior and hyperphagia. Less anger and sadness and an improvement in social behavior were noted during oxytocin treatment when compared with the placebo, as was documented by the participants’ parents. The lack of response in the older group could be related to the small number of older children, a miscalculation of the oxytocin dose, the wrong administration of oxytocin, or that the behavior of older children may be more fixed in their personality and may need a longer duration of treatment to generate a significant effect [24].

The fourth study was a double-blinded, placebo-controlled, crossover trial performed on 24 children with PWS, reported by Miller et al. [25] in 2017. The participants received 16 IU of intranasal oxytocin or placebo for five days, followed by a four-week washout period, and finally an adjustment to the alternative treatment for another five days. The Aberrant Behavior Checklist, Social Responsiveness Scale (SRS-P), Repetitive Behavior Scale- Revised (RBS-R), Hyperphagia Questionnaire, and the Clinical Global Impression (CGI) scale were used to evaluate hyperphagia and other behaviors. The authors reported that the use of oxytocin in children over five days was safe, and a significant improvement was noted in anxiety and self-injurious behavior when compared to the placebo [25].

The fifth study was performed to examine the safety and efficacy of a single dose of oxytocin in infants less than six months of age with PWS. Oxytocin was well-tolerated in infants, with no side effects noted during the seven days of treatment. The effects of oxytocin on oral feeding and social skills in human infants were first reported in this study. Sucking and swallowing were also evaluated before and after oxytocin administration by using the Neonatal Oral–Motor Assessment Scale (NOMAS), video fluoroscopy of swallowing, and the Clinical Global Impression (CGI) scale. The authors reported positive treatment effects on social and feeding behaviors [26].

Damen et al. [27] in 2021 conducted a randomized, double-blind, placebo-controlled cross-over study on the effects of oxytocin versus placebo treatment for three months in children with PWS. The primary outcomes were changes in social behavior and hyperphagia, with differences between males and females and between PWS molecular genetic classes [4]. Forty-six children with PWS were blindly assigned to receive twice-daily intranasal doses of either oxytocin or a placebo during four visits in an outpatient clinic setting. The parents completed questionnaires at all visits. The completed questionnaire forms (e.g., Oxytocin Questionnaire and Dykens Hyperphagia Questionnaire) measured changes in eating or repetitive behaviors using the Repetitive Behavior Scale-Revised (RBS-R) form. According to the Dykens Hyperphagia Questionnaire form, patients who received oxytocin showed a trend of being less hyperphagic. RBS-R scores showed that patients receiving oxytocin had less repetitive behavior. The Oxytocin Questionnaire and Dykens Hyperphagia Questionnaire scores were significantly improved during oxytocin treatment in comparison with placebo treatment in patients with the 15q11-q13 deletion, while patients with maternal disomy 15 showed no differences between oxytocin and placebo treatment. In addition, there were positive effects of oxytocin treatment in PWS males compared to females. The cause for this was not known, but it could be explained by differences in the oxytocin biological system between males and females. Males may be more sensitive to the effect of oxytocin than females; however, more studies are needed. The limitations of this study included the small sample size and the insufficient number of questionnaires used for the assessment of changes in both eating and social behavior [27].

The final study conducted by Hollander et al. [28] in 2021 on children and adolescents with PWS investigated the effect of intranasal oxytocin on hyperphagia and repetitive behaviors. Their double-blind, placebo-controlled, randomized trial was conducted over eight weeks on 23 patients with PWS. The participants received 16 IU/per day and had evaluation visits every two weeks for an eight-week period. The primary outcome measure of food-seeking behavior was assessed by using the Dykens Hyperphagia Questionnaire form. The secondary outcomes were repetitive behaviors, assessed using the Repetitive Behavior Scale-Revised (RBS-R). Significant reductions in hyperphagia and repetitive behaviors were noted across time for the placebo group. There was no reduction for the group that received oxytocin treatment, which could be due to subjective expectation bias from caregivers and clinicians [28]. Overall, the reported studies demonstrated the beneficial effects of oxytocin treatment in PWS, except for the two study trials described. The absence of a positive effect of oxytocin treatment in all studies may reflect the dosage and administration differences and may not indicate that oxytocin does not have a role in treating the PWS phenotype, particularly hyperphagia and behavioral problems. More studies are needed to further understand the role of oxytocin in treating those with PWS.

## 4. Setmelanotide Clinical Trial in PWS

A clinical trial on 40 participants diagnosed with PWS was sponsored by Rhythm Pharmaceuticals (Boston, MA, USA) to treat hyperphagia and obesity. The preliminary results were reported in abstract format at the 2017 PWSA Scientific Conference (Orlando, FL, USA) and are available online [29,30,31]. Studies on Setmelanotide, a melanocortin (MC)-4 receptor agonist that impacts satiety and feeding centers to decrease eating, were undertaken with a once-per-day administration via subcutaneous injection in those with PWS. The proof-of-concept trial included 40 participants diagnosed with PWS (19 males and 21 females) with a mean age of 26.4 years, with an age range of 16 to 25 years; a mean body mass index (BMI) of 39.4 kg/m^2^, with a BMI range of 26.1 to 74.1; and a mean Dykens Hyperphagia Questionnaire (HQ) score of 23.9, with a range of 12 to 45. 

The results of this phase-two study using the MC-4 receptor (MC4R) agonist (Setmelanotide) were obtained in a four-week trial at five centers in the U.S.A. A primary study was randomized as a double-blind comparison of a placebo to three daily doses of Setmelanotide (0.5, 1.5, and 2.5 mg) preceded by a two-week, single-blind placebo run-in time interval. A percent bodyweight change was the primary endpoint, with secondary endpoints including HQ scores; dual-energy x-ray absorptiometry (DEXA) measures of body composition for fat, muscle, and bone; metabolic and laboratory parameters; and safety with tolerability assessments. The mean weight changes at four weeks showed no difference when comparing the Setmelanotide versus the placebo. The mean hyperphagia questionnaire scores demonstrated a small, not-statistically-significant reduction from baseline at the two highest Setmelanotide doses. No changes were observed in the DEXA measurements or laboratory findings. Adverse events included occasional, mild-to-moderate injection-site reactions reported in approximately two-thirds of the participants for both active and placebo administration. Skin and nevi darkening was noted, along with intermittent, spontaneous penile erections. There were no serious adverse events, but one patient discontinued the trial due to injection site reactions. Although the results in PWS were not promising, later studies in other rare, monogenic obesity disorders, such as those with POMC gene mutations or Bardet–Biedel syndrome—a ciliary protein group of genetic disorders—have met with success using this MC4R agonist [32].

## 5. Diazoxide Choline Controlled-Release Clinical Trial in PWS

Diazoxide choline, a new chemical entity, is a benzothiadiazine that acts by stimulating an ion flux through ATP-sensitive K+ channels (KATP). It is the choline salt of diazoxide, and is currently used to treat infants, children, and adults with hyperinsulinemia hypoglycemia. Diazoxide choline controlled-release (DCCR) is diazoxide choline formulated as an oral, once-a-day, extended-release tablet.

The hyperphagia signal in PWS likely occurs due to the dysregulation of neuropeptide Y/Agouti Related Protein/Gamma-aminobutyric Acid (NAG) neurons, which are regulated by leptin via reduction in their excitability [33]. This dysregulation results in marked elevations in the synthesis and secretion of NPY, the most potent endogenous neuropeptide. Leptin’s activation of adenosine triphosphate (ATP)-sensitive potassium channels (K_ATP_) via phosphoinositide-3-kinase (PI3-K) [34,35,36] serves to hyperpolarize the resting membrane potential, which results in a limitation in the release of NPY by these neurons. Depolarizing the resting membrane potential of neurons in the arcuate nucleus (including the NAG neurons) via perfusion with potassium chloride results in the doubling of the NPY release rate, which returns to normal following perfusion [37]. There is strong evidence that the activation of NAG neurons results in insulin resistance and impaired glucose tolerance [38]. Inhibiting these neurons by agonizing the K_ATP_ channel has the potential to improve insulin sensitivity and improve glucose tolerance. Agonizing the K_ATP_ channel in NAG neurons using diazoxide choline is expected to result in reduced NPY secretion. Diazoxide readily crosses the blood–brain barrier [39], and diazoxide choline can be orally administered and can effectively agonize the K_ATP_ channels in the hypothalamic NAG neurons. Therefore, agonizing the K_ATP_ channel in these neurons amplifies the regulatory effects of leptin, reducing the secretion of NPY and likely AgRP and GABA, blunting hyperphagia and impacting obesity. 

This preliminary trial consisted of a single-center, Phase II clinical study including a 1ten-week open-label treatment period followed by a four-week double-blind, placebo-controlled, randomized-withdrawal treatment period conducted at the University of California, Irvine. Patients were initiated on a once-daily oral DCCR dose of approximately 1.5 mg/kg (maximum starting dose of 145 mg) and titrated every two weeks to approximately 2.4 mg/kg, 3.3 mg/kg, 4.2 mg/kg, and 5.1 mg/kg (or to a maximum dose of 507.5 mg, whichever was less) at the discretion of the investigator. Any patient who showed any increase in resting energy expenditure and/or any reduction in hyperphagia from baseline through week six or week eight was designated a responder and was eligible to be randomized in the double-blind treatment period. During the double-blind treatment period, all individuals designated as responders were to be randomized in a 1:1 ratio to either continue active treatment at the dose they were treated at during week eight or to the placebo equivalent of that dose for an additional four weeks. Non-responder patients continued open-label treatment for an additional four weeks. Screening began in June 2014, and the last subject’s visit took place in April 2015. The trial was registered on www.clinicaltrials.gov, identifier NCT02034071. Hyperphagia was measured in the study using a nine-question Modified Dykens Hyperphagia questionnaire, posed to the parent or caregiver, which utilized a two-week recall period. Changes in body fat and lean body mass were measured using dual-energy x-ray absorptiometry (DEXA) at baseline and at the end of the open-label treatment period. Behavioral assessments were also conducted using a questionnaire to assess the presence or absence of 23 PWS-associated behaviors (grouped into four categories) at baseline and at the end of the open-label treatment period. Resting energy expenditure (REE) and respiratory quotient (RQ) were measured by indirect calorimetry. 

The patients enrolled included both males and females and consisted of overweight and obese patients between the ages of 10 and 22 years with genetically confirmed PWS. Fifteen patients were screened, and thirteen subjects were enrolled in the study. Patients treated with growth hormone for at least one year prior to the start of study could be enrolled. 

The open-label treatment period was completed on 11 of the 13 enrolled patients (84.6%). All 11 patients who completed the open-label treatment period were designated as responders and showed improvements in hyperphagia. Most had also shown improvements in REE and were randomized into the double-blind treatment period. All 11 patients who were randomized into the double-blind treatment period completed the period [40]. After two weeks of treatment, there was a significant reduction in the hyperphagia score, and comments by parents were supportive of a marked improvement in hyperphagia. Treatment with DCCR for ten weeks had a significant impact on body composition, including reductions in body fat, increases in lean body mass, and a marked increase in the lean-body-mass to body-fat ratio. However, there was no significant change in weight from baseline to the end of the study. Waist circumference was significantly reduced during the open-label treatment period, suggesting the loss of visceral fat.

## 6. Livoletide Clinical Trial in PWS

Livoletide is an unacylated or inactive ghrelin analogue which works by decreasing the amount of active ghrelin in the brain. Ghrelin is a neuropeptide, produced by the stomach, which directly stimulates eating behavior in the hypothalamus in humans and is reportedly elevated in PWS (e.g., [41]). The ZEPHYR study sponsored by Millendo (Ann Arbor, MI, USA) was a randomized, double-blind, placebo-controlled, pivotal Phase 2b/3 study [42]. The therapy showed promising results in the phase 2a trial, in which daily Livoletide treatment or a placebo were administered via subcutaneous injection for 12 weeks. The Phase 2b study included 158 patients with PWS who were randomized to either (60 µg/kg or 120 µg/kg) receive the Livoletide dosage or the placebo. Livoletide was well-tolerated during this time, and the most-reported side effect was an injection site reaction of mild severity. However, no significant change was found in the Hyperphagia Questionnaire for Clinical Trials (HQ-CT) scores, which measured hunger and food-related behaviors, when compared with placebo. Hence, Livoletide also did not significantly improve hyperphagia and food-related behaviors and had no effect on fat mass, body weight, or waist circumference via DEXA data. As this drug was rigorously studied and the results were negative, it has been suggested that ghrelin may not be a driving force for the hyperphagia observed in PWS patients via this therapeutic route. However, this study may stimulate other gene–protein interactions or pathway analysis to study therapeutic options for treating hyperphagia and obesity in PWS. The company announced that it would discontinue this therapeutic development as a potential treatment model for PWS.

## 7. Cannabinoid Use in PWS

The cannabinoid-1 receptor (CB1R) plays an important role in the regulation of appetitive behavior. CB1Rs are important elements, encoded by genes located on chromosome 6 at bands q14-q15. The receptors are expressed most densely in the brain in areas involved in appetite regulation in the hypothalamus, but they also present in many peripheral tissues such as the liver, adrenal and pituitary glands, adipose tissue, and gonads. Research has shown that the activation of CB1R increases appetite, and the blockade of peripheral CB1R can decrease obesity and metabolic complications. Cannabidiol (CBD) is one of the common, non-psychotropic constituents of the Cannabis plant. It has an antagonist effect on CB1R, and thereby an anti-obesity effect [43]. Its use in PWS to treat hyperphagia progressed to an initial clinical-trial development with subject recruitment; however, the trial did not progress due to unforeseen problems.

## 8. Exenatide Use in PWS

Glucagon-like peptide-1 (GLP-1) is a hormone synthesized from the L- cells of the ileum and colon. It is released in response to food intake, contributing to postprandial glucose regulation. It also augments meal-related insulin secretion from the pancreas. The effect of GLP-1 receptor agonists on weight loss has been studied, with delays in gastric emptying and decreased appetite noted [44]. Exenatide is a GLP-1 receptor agonist, and its use has resulted in persistent weight loss in animals and obese adults [43]. In 2016, Salehi et al. [45] investigated the effect of exenatide treatment in a six-month trial in ten obese, adult patients with PWS. Body weight, body mass index (BMI), truncal fat, appetite measures, and plasma-acylated ghrelin and leptin levels were analyzed. They found no significant effect on body weight, BMI, or truncal adiposity, but did note a significant decrease in appetite scores and eating behavior [45], requiring more testing.

## 9. Transcranial Direct-Current Stimulation (tDCS) Clinical Trial and Startle Response in PWS

Transcranial direct-current stimulation (tDCS) is a safe, painless, and non-invasive technique to modify neuronal and cognitive function in areas of the brain (e.g., [46]). This technique has been undertaken in multiple studies in humans and can stimulate targeted brain regions and networks to increase or decrease cortical excitability in children or adults. In addition, the dorsolateral prefrontal cortex (DLPFC) is involved in the regulation and processing of food craving and motivation accessible by tDCS [46,47,48].

In 2015, Bravo et al. [46] investigated the efficacy of tDCS on the right DLPFC to modulate food craving and hyperphagia in patients with PWS by undertaking a double-blind, sham-controlled multicenter study of tDCS in ten adults with genetically confirmed PWS, eleven adult, obese subjects, and eleven adult, healthy-weight controls. The PWS and obese subjects received five consecutive daily sessions of active or sham tDCS over the right DLPFC, while healthy-weight subjects received a single sham and active tDCS in a cross-over design. Standardized assessments for food cravings, drive, and hyperphagia from self-reporting and caregiver information were obtained over 30 days. The observed baseline differences were robust for severity scores for the Three Factor Eating Questionnaire (TFEQ) and the Dyken’s Hyperphagia Questionnaire (DHQ) for PWS subjects when compared to healthy-weight controls, while the obese subjects were more similar to the healthy-weight controls. Active tDCS sessions in PWS were associated with a significant change from baseline in the TFEQ disinhibition component and for total scores. The participant ratings for the DHQ severity category and total scores were also significant. The reports of food craving using the Food Craving Analogue Scale, provided through self-reporting or by caregivers, supported the hypothesis of primary disturbances in cognitive and emotional aspects and food preoccupation in PWS [46]. 

In 2021, Poje et al. [48] further investigated the effect of tDCS on the DLPFC and hyperphagia in patients with PWS by reporting on the positive effects of brief tDCS sessions on the Go/NoGo task performance involving food and non-food stimuli images. Alterations in N2-brainwave-amplitude were recorded utilizing a skull-cap electroencephalogram apparatus, and PWS molecular genetic class differences before and after tDCS were assessed by event-related potentials (ERPs) in ten adults with PWS. The results indicated a group effect on baseline NoGo N2 amplitude in PWS patients with the 15q11-q13 deletion versus maternal disomy 15, a decrease in NoGo N2 brain amplitude in PWS adults with the deletion versus maternal disomy 15, and a decrease in NoGo N2 amplitude following tDCS. The tDCS approach demonstrated a trend towards a decreased response time. It collectively replicated and expanded prior work, highlighting neurophysiological differences in patients with PWS according to their genetic subtype and demonstrating the feasibility of examining neuromodulatory effects on information processing in individuals with PWS. The study supported the positive effect of tDCS on hyperphagic behavior in patients with PWS [46], requiring more testing.

In 2018, Gabrielli et al. [49] reported on the emotional processing of food and eating behavior in PWS by using startle-response modulation. The startle-eyeblink response is an involuntary reflex activated by the autonomic nervous system in response to sudden or disturbing auditory/visual stimuli. It may be modulated by the emotional valence of concurrently viewed visual stimuli. Gabrielli et al. studied thirteen individuals with PWS and eight healthy controls while viewing standard neutral, negative, positive, and food-derived images. Electromyogram (EMG) recordings of the orbicularis oculi muscle were measured in response to binaural white noise before and after the consumption of a standard 500 kcal meal. Participants reported their pre- and post-meal perceived emotional valence for each image using a one to ten Likert rating scale. Subjective ratings of food images and the urge to eat were significantly higher in PWS than found in controls and did not significantly decline post-meal. Acoustic startle responses detected in PWS were significantly lower than observed in controls under all conditions. Startle responses to food images in PWS were attenuated relative to other picture types, with a potentially abnormal emotional modulation of responses to non-food images, which contrasted with self-reported picture ratings. A stable, positive emotional valence to food images was observed pre-and post-feeding with a sustained urge to consume food in the PWS participants. Researchers concluded that the emotional processing measured using startle-modulation-responsive, non-food images was abnormal in PWS, which may reflect their unique features, such as hypotonia or increased fat mass and distribution, possibly impacting skin conductivity. It may also reflect an autonomic-nervous-system derangement in PWS that requires more testing [49]. 

## 10. Surgical Management of Obesity in PWS

The role of bariatric surgery in patients with PWS is controversial. Alqahtani et al. [50] and Fong et al. [51] reported decreased food seeking and a reduction in body weight and comorbidities such as obstructive sleep apnea, hypertension, and diabetes mellitus in PWS patients after laparoscopic sleeve gastrectomy. By contrast, Liu et al. [52] reported the rebound of symptoms and the development of obesity complications four to five years following bariatric surgery, which was described as ineffective in producing sustainable weight loss. 

Scheimann et al. [53] reported a retrospective review on 60 published cases of PWS patients who underwent bariatric surgery. They found a variety of postoperative complications in PWS patients after bariatric surgery. Patients with PWS have special medical problems, such as a higher degree of insulin sensitivity, growth-hormone deficiency, hyperlipidemia, a decreased ability to vomit, and abnormal eating behavior with hyperphagia; therefore, a high potential for the development of gastric dilation/necrosis exists. These factors suggest that patients with PWS may have a higher potential risk of bariatric surgical complications than those seen in obese individuals without PWS. After jejunoileal bypass, one patient died. Another experienced wound infection and deep vein thrombosis. Furthermore, the patients who underwent gastric bypass with PWS found that one patient died, and two patients underwent splenectomy at the time of bariatric surgery. Most of the PWS patients who underwent gastroplasty reported a loss of weight, but this was followed by rebound weight gain. However, one PWS patient underwent laparoscopic, silicone gastric banding and died 45 days post-operatively [53]. Additionally, DePeppo et al. [54] reported a significant improvement in BMI after endoscopic intragastric balloon (BIB) placements in 12 patients with PWS, but two patients died from gastric perforation. Two other PWS patients developed severe, symptomatic gastric distension associated with food consumption. Although there are a small number of reported case series in PWS, there appears to be little justification for subjecting PWS patients to the potential high risk of complications from bariatric surgery (e.g., [53]).

## 11. Conclusions

As most individuals with PWS begin marked food seeking and hyperphagia during early childhood and often develop extreme obesity over time if uncontrolled, a safe and effective treatment is warranted to control both hyperphagia and obesity in PWS. Hence, several agents are currently in the clinical-trial stage or have been studied for treating hyperphagia and obesity in PWS patients as described. No specific differences were noted in drug responses in the individual clinical trials related to the PWS molecular genetic classes identified in the enrolled participants. Other drugs which impact features seen in PWS includes the antiepileptic drug topiramate, which showed a potential effect on disordered eating in several small studies in PWS patients [55,56]. It is also recommended for the treatment of skin-picking, a common obsessive-compulsive manifestation observed in individuals with PWS [57]. Furthermore, Holland et al. [58] recently investigated the efficacy of transcutaneous vagus nerve stimulation (t-VNS) for the treatment of temper outbursts and related behavioral findings in PWS, including hyperphagia; five individuals were studied. They found a significant decrease in temper outbursts in four out of the five PWS participants, with an improvement in emotional control, responses to interventions, and an increased ability to control and manage outburst-stimulating situations. However, they did not observe any reduction in hyperphagia [58].

Various drugs and devices investigated in clinical trials are discussed in this review and are summarized in Table 1, including beloranib, setmelanotide, DCCR, unacylated ghrelin analogue, oxytocin or related compounds, glucagon-like peptide 1 receptor agonists, transcranial direct-current stimulation, and transcutaneous vagus nerve stimulation. Other agents have been used, proposed, or are under development such as tesofensine/metoprolol, cannabinoids, topiramate, and histamine-related agents for the treatment of PWS findings that include hyperphagia and obesity [28,55,56,59,60,61]. Medical devices for treating hyperphagia and modulating eating behavior in PWS have shown promise. These include transcranial direct-current stimulation and transcutaneous vagus nerve stimulation in individuals with PWS, as well as generated information relating to brain regions and function impacted by the PWS diagnosis which require more testing. No specific differences were noted in response to the individual clinical-trial protocols or therapeutic agents related to the PWS molecular genetic classes identified in the PWS participants enrolled in the clinical trials [23,40,46,47,48,49,58]. 

Two of the trials that we reviewed hold promise for impacting hyperphagia and obesity in PWS. The reported success for the use of beloranib for weight loss and decreased appetite would have been of interest for continued testing; however, this trial was discontinued due to reported deaths. The second trial used DCCR tablets to treat hyperphagia and obesity and reported changes in both body composition measures and hyperphagia in a small number of subjects with PWS. More detailed studies are currently under study and review. Oxytocin trials have shown mixed results in treating PWS; this may be related to identifying the correct dose needed for use and may depend on the age of participants. Additionally, more research is needed to identify more objective measures for determining the level of hyperphagia. For example, the startle response and food-image processing in Prader–Willi syndrome reported by Gabrielli et al. [49] before and after consuming a standard meal may prove useful as an objective measure of hyperphagia and food-driven responses. More research is needed to identify objective measures of hyperphagia instead of using the subjective questionnaire forms that current exist for clinical trial use. Body composition measures using DEXA, weight, circumference, and body mass index (BMI) are the currently available objective data for studying obesity in clinical trials.

New pharmacotherapies aimed at controlling hyperphagia and appetite behavior in PWS patients are of interest for this patient population, with potential direct applications in the treatment of non-syndromic or exogenous obesity, which is on a significant rise worldwide and contributes to growing morbidity and mortality and healthcare costs. The identification of new mechanisms of the cause of hyperphagia and weight gain in patients with PWS, based on advances in genetic technology, bioinformatics, and gene variant testing, will be important. Computational biology to identify gene–gene–protein interactions and biological–molecular processes with pathways will lead to a better understanding of the cause and mechanism of other rare, obesity-related disorders (e.g., [62]). The recently reviewed research regarding the genetics of obesity and the newly acquired information about the role of genes [14,62,63] will yield promising new molecular targets for potentially novel pharmaceutical agents. 

## Figures and Tables

**Figure 1 ijms-24-02150-f001:**
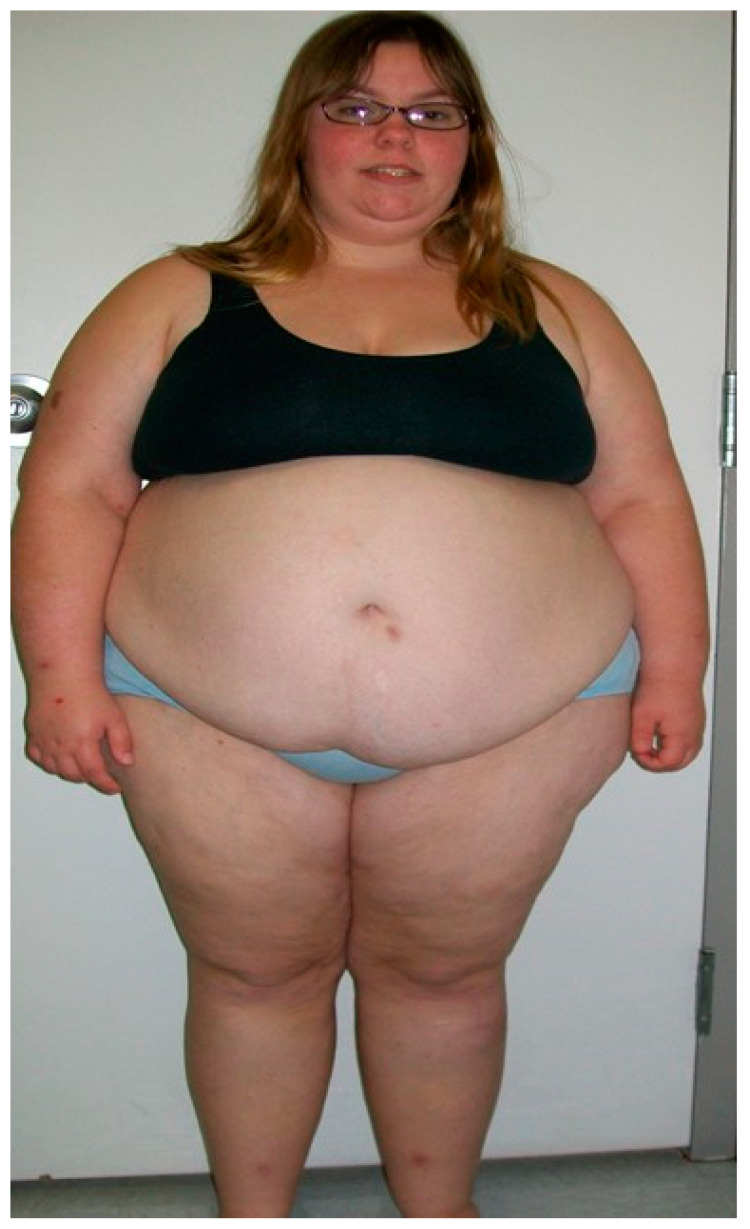
Frontal view of a 16-year-old female with Prader–Willi syndrome due to maternal disomy 15. (Modified from Mahmoud et al. [14]).

**Table 1 ijms-24-02150-t001:** Summary of drugs and medical devices used in PWS clinical trials.

Mechanism of Action	Studies Reviewed	Age Range	Reason Chosen for Treatment
**Beloranib** Beloranib inhibits methionine aminopeptidase 2 (MetAP2) by removing methionine residue from proteins, impacting fat metabolism and adipocyte size in animal models.	McCandless et al. [15]	12–65 years	Inhibitors of MetAP2 were found to reduce food intake, affect adipose tissue, and reduce fat synthesis with weight loss in humans.
**Oxytocin** Oxytocin is a neuropeptide hormone produced in the brain that plays an important role in social interactions and skills, food intake, anxiety, energy expenditure, and body-weight regulation.	Tauber et al. [22] Einfeld et al. [23] Kuppens et al. [24] Miller et al. [25] Tauber et al. [26] Damen at al. [27] Hollander et al. [28]	18.7–43.6 years >12 years 6–14 years 5–11 years <6 months 3–11 years 5–18 years	Patients with PWS have been reported to have decreased oxytocin-producing neurons. This deficiency could be related to their inability to control their emotions, with poor social adjustment and food intake.
**Setmelanotide** Setmelanotide is a melanocortin (MC)-4 receptor agonist that impacts satiety and feeding to decrease eating.	Rhythm Pharmaceuticals [29,30,31]	16–25 years	Patients with PWS begin marked food seeking and hyperphagia during early childhood and develop extreme obesity over time if not externally controlled.
**Diazoxide choline controlled release (DCCR)** DCCR is a benzothiadiazine that acts by stimulating ion flux through ATP-sensitive K+ channels used to treat infants, children, and adults with hyperinsulinemia hypoglycemia.	Kimonis et al. [40]	10–22 years	Hyperphagia in PWS relates to dysregulation of neuropeptide Y/Agouti Related Protein/Gamma-aminobutyric Acid (NAG) neurons, which are regulated by leptin via the reduction of their excitability. This dysregulation results in marked elevations in the synthesis and secretion of NPY, the most potent endogenous neuropeptide. Leptin’s activation of adenosine triphosphate (ATP)-sensitive potassium channels (K_ATP_) via phosphoinositide-3-kinase (PI3-K) serves to hyperpolarize the resting membrane potential, resulting in a limitation of the release of NPY by these neurons, thus blunting the hyperphagia signal.
**Livoletide** Livoletide is an inactive ghrelin analogue which works by decreasing the amount of the active form of ghrelin in the brain. Ghrelin is a neuropeptide produced by the stomach which directly stimulates eating behavior in the hypothalamus in humans.	Millendo Therapeutics SAS [42]	8–65 years	Patients with PWS have elevated ghrelin levels.
**Exenatide** Glucagon-like peptide-1 (GLP-1) is a hormone synthesized from L- cells of the ileum and colon and released in response to food intake. GLP-1 receptor agonists such as Exenatide affect weight loss in the form of a delay in gastric emptying and decreased appetite.	Salehi et al. [45]	13–25 years	Exenatide is a GLP-1 receptor agonist and its use has resulted in persistent weight loss in animals and obese adults.
**Transcranial direct-current stimulation (tDCS)** Transcranial direct-current stimulation (tDCS) is a safe, painless, and non-invasive technique to modify neuronal and cognitive function in areas of the brain to help modulate food craving.	Bravo et al. [46] Poje et al. [48] Gabrielli et al. [49]	18–64 years 19–44 years 16–65 years	The dorsolateral prefrontal cortex (DLPFC) is involved in the regulation and processing of food craving and motivation in humans.

## Data Availability

The data supporting the reported material can be obtained upon request.

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
