# Peer review of "Clinical Trials in Prader–Willi Syndrome: A Review"

_ijms, 2023, doi:10.3390/ijms24032150_

Round 1

Reviewer 1 Report

This is an exhaustive review of clinical trials in PWS which will be appreciated generally and especially by people interested in this rare syndrome and related hyperphagia-obesity disorders. My major criticism has to do with its length, that I consider excessive. The description of some trials, like for instance the Beloranib trial and the Rhythm sponsored study, is way too long and detailed and it should be shortened, to convey the essence of these trials in a synthetic manner. One other point is that the prose is not always smooth and clear. Just a few examples: In lines 16-17 there is a redundancy that does not sound well. The phrase in lines 53-55 is clearly missing something to be comprehensible. Again in line 55 there is mention of the death rate, that makes no sense without reference to age. The phrase in lines 73-76 has a structure that makes it difficult to read. In line 202 there must be one word missing. The phrase in lines 484-487 does not stand as it is. I strongly recommend that the senior authors re-read the MS carefully and make the necessary corrections.  One last point. Since the authors are recognized experts in the field, I recommend that in the Conclusions they present to the reader their own overall opinion and judgement on the reported trials, and to the families a guideline to orient themselves in a tangle of treatment options.

Author Response

           December 19, 2022

Prof. Dr. Maurizio Battino

Editor-in-Chief

Department of Odontostomatologic and Specialized Clinical Sciences,

Sez-Biochimica, Faculty of Medicine, Università Politecnica delle Marche,

Via Ranieri 65, 60100 Ancona, Italy

We thank you for permitting us to make revisions to further strengthen our manuscript for publication. We appreciate the detailed critiques of the two reviewers and have provided a point-by-point response to the concerns in the cover letter and in the revised manuscript

Response: Beloranib trial and Rhythm trial were summarized and shortened as requested.

One other point is that the prose is not always smooth and clear. Just a few examples: In lines 16-17 there is a redundancy that does not sound well.

Response: The sentence was corrected.

The phrase in lines 53-55 is clearly missing something to be comprehensible.

Response: The sentence was corrected to be comprehensible.

Again, in line 55 there is mention of the death rate, that makes no sense without reference to age.

Response: This sentence was deleted.

 The phrase in lines 73-76 has a structure that makes it difficult to read.

Response: These sentences were deleted.

In line 202 there must be one word missing. The phrase in lines 484-487 does not stand as it is. I strongly recommend that the senior authors re-read the MS carefully and make the necessary corrections. 

Response: The manuscript was revised by the senior authors and significant changes were done in the text and length of manuscript to address the reviewers' comments.

One last point. Since the authors are recognized experts in the field, I recommend that in the Conclusions they present to the reader their own overall opinion and judgement on the reported trials, and to the families a guideline to orient themselves in a tangle of treatment options.

Response:

A new paragraph was added by the senior authors who are expertise in PWS in the conclusion representing their overall opinion about the reported clinical trials.

Yours Sincerely, 

Merlin G. Butler, MD, PhD,

Department of Psychiatry and Behavioral Sciences

University of Kansas Medical Center

3901 Rainbow Blvd., MS 4015

Kansas City, KS, 66160, USA

Reviewer 2 Report

This is a very broad and detailed review on the clinical trials performed in PWS so far which will be very helpfull for readers seeking information both on the outcome and the metodical quality of the investigations in the field.

I have only a few minor comments:

1. In the abstracts the two sentences from line 14 - 19 should be checked for language and meaning.

2. most chapter headlines contain the investigated drug or other therapeutic measures. The headlines of chapter 2 and 4, however, contain the names of the sponsoring company. Why is that so? This should be changed.

3. In  table 1 first line I recommend to put the sponsor in brackets and the mane of the drug first. A fragment of this tabel is repaeated on page 13, probably a technical problem.

Author Response

Prof. Dr. Maurizio Battino

Editor-in-Chief

Department of Odontostomatologic and Specialized Clinical Sciences,

Sez-Biochimica, Faculty of Medicine, Università Politecnica delle Marche,

Via Ranieri 65, 60100 Ancona, Italy

We thank you for permitting us to make revisions to further strengthen our manuscript for publication. We appreciate the detailed critiques of the two reviewers and have provided a point-by-point response to the concerns in the cover letter and in the revised manuscript.

This is a very broad and detailed review on the clinical trials performed in PWS so far which will be very helpfull for readers seeking information both on the outcome and the metodical quality of the investigations in the field.

Response: Thank you for the positive feedback.

I have only a few minor comments:

  1. In the abstracts the two sentences from line 14 - 19 should be checked for language and meaning.

Response: The sentences were corrected.

  1. most chapter headlines contain the investigated drug or other therapeutic measures. The headlines of chapter 2 and 4, however, contain the names of the sponsoring company. Why is that so? This should be changed.

Response: We removed the names of the sponsoring company and used the drug names.

  1. In  table 1 first line I recommend to put the sponsor in brackets and the mane of the drug first. A fragment of this tabel is repaeated on page 13, probably a technical problem.

Response: We removed the names of the sponsoring company, and the table was corrected.

Yours Sincerely, 

Merlin G. Butler, MD, PhD,

Department of Psychiatry and Behavioral Sciences

University of Kansas Medical Center

3901 Rainbow Blvd., MS 4015

Kansas City, KS, 66160, USA
